# Prevalence and Risk Factors of Deep Venous Thrombosis in Intensive Inpatient Neurorehabilitation Unit

**DOI:** 10.3390/healthcare12090936

**Published:** 2024-05-01

**Authors:** Maria Elena Pugliese, Riccardo Battaglia, Maria Ursino, Lucia Francesca Lucca, Maria Quintieri, Martina Vatrano, Paolo Tonin, Antonio Cerasa

**Affiliations:** 1Intensive Rehabilitation Unit, S’Anna Institute, 88900 Crotone, Italy; dr.riccardo.battaglia@gmail.com (R.B.); m.ursino@isakr.it (M.U.); l.lucca@istitutosantanna.it (L.F.L.); m.quintieri@isakr.it (M.Q.); m.vatrano@isakr.it (M.V.); patonin18@gmail.com (P.T.); 2Institute for Biomedical Research and Innovation (IRIB), National Research Council of Italy, 98164 Messina, Italy; 3Pharmacotechnology Documentation and Transfer Unit, Preclinical and Translational Pharmacology, Department of Pharmacy, Health Science and Nutrition, University of Calabria, 87036 Rende, Italy

**Keywords:** deep venous thrombosis, rehabilitation, D-dimer, compressive ultrasonography, severe brain injury, stroke

## Abstract

Venous thromboembolism (VTE) (deep vein thrombosis and its complication, pulmonary embolism) is a major cause of morbidity and mortality in hospitalized patients and about 7% of these cases are due to immobility secondary to a neurological impairment. Acquired brain injury (ABI) has also been recognized as one of the main risk factors for VTE. Numerous epidemiological studies have been conducted to assess the risk factors for VTE in institutionalized polytrauma patients, although there is a lack of information about neurorehabilitation wards. Since VTE is often undiagnosed, this prospective study aimed to determine the prevalence and clinical characteristics of lower-limb deep venous thrombosis (DVT) in ABI patients at neurorehabilitation admission. Methods: ABI patients were screened for DVT on admission to the intensive rehabilitation unit (IRU) with compression ultrasonography and basal D-dimer assay and were daily clinically monitored until discharge. A total of 127 consecutive ABI patients (mean age: 60.1 ± 17.6 years; 63% male; time from event: 30.9 ± 22.1 days; rehabilitation time in IRU: 84.6 ± 58.4 days) were enrolled. Results: On admission to the IRU, the DVT prevalence was about 8.6%. The mean D-dimer level in patients with DVT was significantly higher than in patients without DVT (6 ± 0.9 vs. 1.97 ± 1.61, *p*-value = 0.0001). ABI patients with DVT did not show any significant clinical characteristics with respect to ABI without DVT, although a prevalence of hemorrhagic strokes and patients originating from the Intensive Care Unit and Neurosurgery ward was revealed. During the rehabilitation period, patients with DVT showed a significant difference in pharmacological DVT prophylaxis (high prevalence of nadroparin with 27.3% vs. 1.7%, *p*-value = 0.04) and a prevalence of transfers in critical awards (36% versus 9.5% of patients without DVT, *p*-value = 0.05). The mortality rate was similar in the two groups. Conclusions: Our research offers a more comprehensive view of the clinical development of DVT patients and confirms the prevalence rate of DVT in ABI patients as determined upon IRU admission. According to our findings, screening these individuals regularly at the time of rehabilitation admission may help identify asymptomatic DVT quickly and initiate the proper treatment to avoid potentially fatal consequences. However, to avoid time-consuming general ultrasonography observation, a more precise selection of patients entering the rehabilitation ward is required.

## 1. Introduction

The third most common acute cardiovascular condition worldwide, after myocardial infarction and stroke, is venous thromboembolism (VTE) [1], which can present clinically as deep vein thrombosis (DVT) or pulmonary embolism (PE) [2]. VTE can be fatal for medical inpatients, although there is evidence that pharmacologic VTE prophylaxis can effectively prevent VTE if started 24 to 72 h after admission [3], decreasing symptomatic PE by 58%, fatal PE by 64%, and symptomatic DVT by 53% [4]. However, roughly only 50% of qualified hospitalized patients receive the recommended thromboprophylaxis.

DVT and PE are generally acknowledged as serious side effects in acquired brain injury (ABI) patients. According to earlier research, polytrauma patients may have a lower-extremity DVT incidence of up to 20% [5,6,7], and DVT is also the leading cause of death for individuals who survive their initial traumatic injuries [7]. One of the main risk factors for VTE is hospitalization. Indeed, Heit and colleagues [8] demonstrated that 48% of the cases of VTE in the community could be attributed to hospitalization for surgery (24%) or for medical illness (22%). In chronic brain injury patients, the incidence of VTE increases [9]. Indeed, because prolonged immobility causes venous stasis and decreased blood flow, it is frequently associated with an increased risk of VTE [8,9,10]. Pottier and colleagues [11] performed a meta-analytic evaluation of epidemiological studies investigating the incidence of VTE in institutionalized patients, demonstrating that prolonged immobility (>3 days) is associated with a two- to three-fold increase in VTE risk.

DVT is more prevalent in traumatic brain injury patients but it has also been reported in individuals with other neurological conditions like spinal injuries, stroke, and Guillain–Barre syndrome that are characterized by acute paralysis [12,13,14,15,16], whereas it is less common in patients with neurodegenerative diseases [17]. The strict correlation between the time elapsed since the traumatic event and the rate at which immobility begins has been demonstrated to influence the occurrence of DVT in several studies [18,19]. For instance, Engbergs et al. [20] demonstrated that within two weeks of being released from the hospital, there was a fifteen-fold greater risk of thrombosis. Over a period of three months, thrombosis risk was linked to minor leg injuries, plaster casts, surgery fractures, and temporary immobilization at home. For in-hospital immobility, the population-attributable risks were 27%, and for immobility outside of hospitals, they were 15%. Again, Sartori and colleagues [21], evaluating 252 acutely ill medical inpatients, suggested that just 3 days of immobilization should be considered a risk factor for DVT. 

Despite many epidemiological investigations evaluating the prevalence and risk factors of DVT in ambulatory or hospitalized cohorts, few prospective studies have been conducted in the post-acute rehabilitation phase. DVT is a significant contributor to morbidity and mortality in individuals who have had significant orthopedic surgery, severe trauma, or an acute stroke [22]. Given the potentially fatal consequences of undetected DVT after failed prophylaxis, some clinicians routinely screen patients with TBI who are asymptomatic by using venous doppler ultrasound (VDU). However, VDU is costly and may have limited sensitivity for asymptomatic DVT. There is no consensus regarding appropriate screening, prophylaxis, or treatment during acute rehabilitation [23,24]. For these reasons, in this study, we aim to evaluate DVT’s local prevalence in asymptomatic ABI patients on admission to an intensive rehabilitation unit (IRU) by using a combination of VDU and D-dimer dosing, and to study DVT patient outcomes.

## 2. Methods

### 2.1. Clinical Enrollment

ABI patients were consecutively admitted to the IRU of the Institute S. Anna (Crotone, Italy) between June 2021 and May 2022. We enrolled only those who met the following inclusion criteria: (1) age ≥ 18 years; (2) acute brain injury, including trauma, hemorrhagic or ischemic stroke, spinal cord injury, anoxia, polyneuritis, or central nervous system infections; (3) written informed consent. The exclusion criteria were as follows: (1) acute brain injury > 90 days; (2) pre-existing severe disability; (3) diagnosis of DVT in an acute setting before rehabilitation admission; (4) neurodegenerative diseases (i.e., Parkinson’s Disease). All patients were transferred directly from the hospital after the medical and neurosurgery complications had been stabilized. The departments transferring patients more frequently were the Intensive Care Unit, Neurosurgery, Neurology, Geriatric, and the Internal Medicine ward. The data from the acute hospital ICU were retrieved from patient files. 

The present study was carried out following the rules of the Declaration of Helsinki and was approved by the local ethics committee of “Regione Calabria Comitato Etico Sezione Area Centro”, n.320, 21 December 2017. Written informed consent was obtained from the patients’ authorized representatives before study enrollment. 

### 2.2. Procedure and Clinical Evaluations 

This was a monocentric prospective observational study. On admission to the neurorehabilitation unit, all patients underwent a detailed anamnesis and clinical evaluation. Specifically, demographic data (sex, age, and body mass index), comorbidities, risk factors for VTE (i.e., SEPSI or fractures) and DVT-related clinical assessment data were collected. 

Diagnosis of DVT was accomplished using an integrated clinical, biochemical, and instrumental evaluation. The DVT screening protocol consisted of the following:An evaluation of DVT clinical signs, such as leg asymmetry with unilateral edema, erythema, and pain, and the presence of Buer and Homans signs.A single quantitative D-dimer assay performed within 24 h of rehabilitation admission. The LIATEST D-dimer assay, which is an immuno-turbidimetric quantitative assay method based on a latex microparticle agglutination test, was used, with a positive D-dimer test for values > 0.5 g/mL [25,26].A compressive leg ultrasound performed on all eligible patients admitted to the rehabilitation unit, irrespective of the D-dimer value (normal value < 0.5 g/mL) [27]. All venous duplex ultrasonography studies were performed by the same radiographer with specific experience and reported by trained radiologists using a Canon Aplio 300 ultrasound machine during the entire study period. A compressive ultrasound was performed on each patient within 72 h of admission. The presence of DVT was determined by a positive venous duplex ultrasonography result [28].

Clinical assessment on admission also included the following: (A)A motor functional evaluation with the Barthel Index.(B)A thrombosis/hemorrhagic risk evaluation with PADUA scores including the improve bleeding risk prediction score, improve associative score for VTE, and improve DD scale [29,30].(C)A laboratory screening protocol on admission included complete blood count, levels of creatinine, azotemia, electrolytes, and reactive C protein (RCP), erythrocyte sedimentation rate (ESR), PT, and aPTT.

Patients who had DVT evidence (DVT+) started anticoagulation therapy, if clinically feasible. In addition, the treatment protocol included highly compressive (23–32 mmHg) monolateral or bilateral socks and early mobilization. Throughout their entire rehabilitation stay, each DVT-negative enrolled patient had a daily clinical evaluation to check the appearance of DVT signs/symptoms (Figure 1). In cases of clinical suspicion, D-dimer testing and compressive leg ultrasound were performed. The diagnosis of PE was made on clinical grounds and investigated accordingly.

### 2.3. Statistical Analysis 

Statistical analysis was performed using SPSS v26 package for Windows (Statistical Package for Social Sciences; www.spss.it (accessed on 1 January 2024); Chicago, IL, USA). Assumptions for normality were tested for all continuous variables. Normality was tested using the Kolmogorov–Smirnov test. Chi2 was used to analyze variations in the distribution of clinical (i.e., risk factors and etiology) and demographic (i.e., gender) characteristics. Both the parametric statistical analysis (unpaired *t*-test) and the non-parametric statistical analysis (Mann–Whitney U-tests and the Wilcoxon signed-rank test) were used for other variables. A *p* < 0.05 cut-off was considered statistically significant for every test.

## 3. Results

Table 1 reported the demographical and clinical information acquired on admission to the IRU. The prevalence of DVT was 8.6%, with only two patients showing clinical signs of DVT (1.45%). ABI patients with DVT tended to be more hemorrhagic and came from the ICU/Neurosurgery ward (Figure 2), although a significant difference with respect to other patients was not revealed (*p*-value = 0.36 and 0.21, respectively, for etiology and origin ward). Similarly, no significant risk factors were detected also considering additional clinical parameters such as thrombophilia, active cancer, estrogen contraceptive therapy, cardiac insufficiency, previous stroke, hypertension, smoking habit, obesity, and pressure wounds. 

ABI patients with DVT are characterized by higher values in the D-dimer assay (6 ± 0.9) and PCR test (5.5 ± 4.2) with respect to ABI patients without DVT (1.97 ± 1.61 and 3.31 ± 3.8; *p*-values = 0.0001 and = 0.03; respectively, for D-dimer assay and PCR). The totality of patients (*n* = 11) diagnosed with DVT by means of color doppler ultrasonography had positive D-dimer tests on admission (sensibility: 100%). No false negative D-dimer result was reported (negative predictive value (NPV): 100%). A total number of 105 patients with a positive D-dimer value were negative upon ultrasound examination (sensibility: 10.25%), with only 12 patients with a negative D-dimer value and negative ultrasound (positive predictive value (PPV): 9.48%). 

Concerning medication, the totality of DVT-positive patients received VTE prophylaxis on rehabilitation admission. On admission, all DVT patients received anticoagulation medication; however, around 10% (n = 11) had full-dose anticoagulation for atrial fibrillation, and the rest of the patients received EBPM or fondaparinux for DVT prophylaxis. Overall, the pharmacological treatment for DVT+ patients differed from that of DVT− patients since they were more frequently prescribed nadroparin at 0.3 mL for thrombosis prophylaxis (27.3% vs. 1.7%, *p* value = 0.04) (Table 2).

During the patients’ rehabilitation stay, no additional cases of DVT were detected. At discharge, a general clinical worsening was detected in ABI patients with DVT, with a prevalence of transfer to critical awards, while no differences were observed in mortality rate and length of stay (Table 3). 

## 4. Discussion

There are various spectrums of DVT and PE, from asymptomatic to cardiopulmonary dysfunction. Neurorehabilitation patients may under-report symptoms of DVT because of cognitive impairment, aphasia, neglect, or altered conscious states [31]. Consequently, thromboembolism is often underdiagnosed in rehabilitation settings. Early detection is essential because DVT therapy, with the aim of thrombus resolution, can save lives. Clinical PE occurs in 26% to 67% of untreated proximal DVT patients and is associated with an 11% to 23% rate of mortality. If treated, these numbers decrease to 5% and 1%, respectively [32]. A leg duplex ultrasound screening conducted upon IRU admission in 26% of ABI patients may require repeat duplex examinations or anticoagulation, changing the course of treatment. [33]. 

In this prospective study, we evaluated the prevalence of DVT symptoms in a large population of ABI patients and the clinical outcome. For the screening of DVT, we combined leg duplex ultrasound with D-dimer quantification and included them in the diagnostic algorithm of “Guidelines for the Diagnosis, Treatment and Prevention of Pulmonary Thromboembolism and Deep Vein Thrombosis” [34,35]. Our findings showed that the overall incidence of thromboembolism in rehabilitation patients on admission after acute brain injury was 8.6%, with only two patients with clinical signs of DVT (1.45%). These results are similar to those of previous studies of patients with ABI in neurorehabilitation settings that reported incidences ranging from 8.5 to 18% when asymptomatic screening was employed vs. only 1.6% for patients with symptomatic DVT [22,29]. In agreement with previous studies, D-dimer test sensitivity and NPV were 100% when using a cut-off value of 0.5 mL/L [36,37]. Notably, the D-dimer value can rise in several circumstances, including infections, pressure ulcers, inflammation/trauma, and post-surgery [38], which is more prevalent in patients with ABI undertaking neurorehabilitation. Therefore, because of its low specificity and positive predictive value, the D-dimer test is not advised for DTV rule-in. In our sample, the PPV (10.25%) and specificity (9.48%) were both extremely low. Again, an optimized cut-off value of 2.5 mg/L increased specificity to 75% for the diagnosis of deep vein thrombosis, at the cost of a modest loss in sensitivity (81%). In agreement with Akman et al.’s [39] findings, the mean D-dimer level in our DVT+ patients was considerably higher than the mean level in patients without DVT (6 ± 0.9 vs. 1.97 ± 1.61, *p*-values = 0.0001). Acute brain injury patients with DVT were also characterized by higher values in the PCR test (5.5 ± 4.2) with respect to ABI patients without DVT (3.31 ± 3.8) (*p*-values 0.03). This result is consistent with a previous case-control study that examined inflammatory markers and found that DVT patients had considerably higher levels of inflammatory markers—like CRP—compared to controls [40].

The second main finding of this study refers to the clinical course of ABI patients with DVT. The 30-day mortality rate for DVT patients not receiving anticoagulation exceeds 3%, and the risk of death increases ten-fold in patients who develop PE [41]. In our cohort, the death rate was the same for both groups. DVT+ patients tended to require more urgent transfers to crucial wards for surgical or medical difficulties (*p*-level = 0.058), thus confirming the importance of early diagnosis in reducing critical clinical complications. Furthermore, a DVT diagnosis can also be regarded as a clinician alert since it indicates a subset of complicated patients who have a high risk of internal problems.

At the pharmacological level, no significant risk factors were identified, except the use of seleparine at 0.3 mL as DVT prophylaxis in 27% of DVT-positive patients compared to 1.7% of the DVT-negative group (*p* value = 0.04), where enoxaparin 4000 UI was more frequently used (65%). This information is in line with the results of a randomized experiment [42], which compared enoxaparin 4000 UI with nadroparin at 0.3 mL in terms of preventing venous thromboembolism. While all patients diagnosed with DVT were receiving medical prophylaxis at the time of admission, 90% of patients who tested negative for DVT started receiving it before admission to rehabilitation. While this result may seem contradictory, it should be remembered that eleven DVT− patients were treated for atrial fibrillation and so received full doses of direct oral anticoagulants. Again, no differences were noticed between the two groups regarding the length of stay in neurorehabilitation and functional impairment measured with the Barthel Index on admission and at discharge. The evaluation of risk scores for thrombosis and hemorrhagic bleeding, such as the Padua score, improve associative score for DVT, improve DD score, and improve bleeding score, revealed a population with extremely high risk for both thrombosis and bleeding, with no discernible differences between patients who were positive and negative for DVT. Finally, ABI patients with DVT tended to be more hemorrhagic and came from the ICU/Neurosurgery ward, although a significant difference with other patients was not revealed. 

## 5. Conclusions

DVT is a frequent complication in neurologic patients entering neurorehabilitation. It is difficult to identify because it is often asymptomatic. Few studies have assessed the results of monitoring the prevalence of DVT in rehabilitation, even though it is notably linked to substantial morbidity in ABI patients. According to our findings, a routine screening using compressive leg ultrasonography and the D-dimer assay at the time of rehabilitation admission can help identify patients with asymptomatic DVT early on and treat them appropriately, potentially improving their prognosis. We advocate the use of ultrasound at the time of neurorehabilitation patient admission because it is a quick, safe, and readily available diagnostic procedure. Nevertheless, a stratified screening program based on an individual patient’s risk has merit. To lower the risk of life-threatening complications and ascertain the incidence of DVT during a rehabilitation stay, more research is required to evaluate the optimal screening approach, saving resources without increasing patient risk. Since it has been demonstrated that patients who developed VTE are more likely to have undergone placement of invasive intracranial monitoring or neurosurgical intervention compared to their counterparts who did not develop VTE [43], in this study, we only reported the number of patients requiring more urgent transfers to crucial areas for surgical or medical difficulties. Therefore, it remains uncertain which repeated medical/neurosurgical interventions were required for DVT+ patients during the neurorehabilitation stay. Next, it remains to be established which risk factors are associated with clinical outcomes in ABI patients with VTE. In agreement with Jacob et al. [44], while early VTE prophylaxis and low molecular weight heparin compared to heparin were found to be protective factors, penetrating injury mechanism, increasing age, male gender, obesity, tachycardia, neurosurgical procedures (craniectomy/craniotomy), and pre-existing hypertension were linked to an increased risk of VTE. In our study, the lack of a regression statistical model did not allow us to evaluate which demographical, clinical, and biological features could be considered a risk factor for VTE in ABI patients. Indeed, small sample sizes (n = 11) can lead to unstable parameter estimates. However, the prevalence of well-known clinical risk factors for DVT was not different between the two groups (Table 1).

Notwithstanding these limitations, our research emphasizes the importance of monitoring the occurrence of DVT in clinical neurorehabilitation units. The importance of these clinical symptoms is supported by the high frequency on admission of hemorrhagic strokes and of patients coming from the Intensive Care Unit and Neurosurgery wards, as well as the greater number of transfers to critical awards throughout rehabilitation stays in DVT+ patients. Our findings could provide valuable knowledge for healthcare professionals, enabling the development of tailored preventive strategies [45], early detection methods [46], and effective management protocols to reduce the burden of DVT during the neurorehabilitation period.

## Figures and Tables

**Figure 1 healthcare-12-00936-f001:**
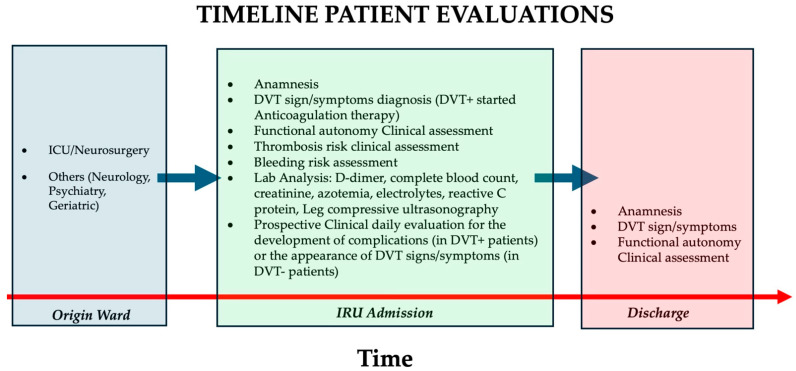
Procedural phases of patient evaluations from origin ward (blue box), during the neurorehabilitation stay in the IRU (green box) and at discharge (rose box). Clinical, laboratory, and ultrasonography evaluations were performed on admission to determine the prevalence of ABI patients with (DVT+) or without (DVT−) lower-limb deep venous thrombosis. Before discharge, a daily clinical evaluation was carried out to assess the development of new DVT symptoms in the enrolled ABI population or the occurrence of complications in DVT+ patients.

**Figure 2 healthcare-12-00936-f002:**
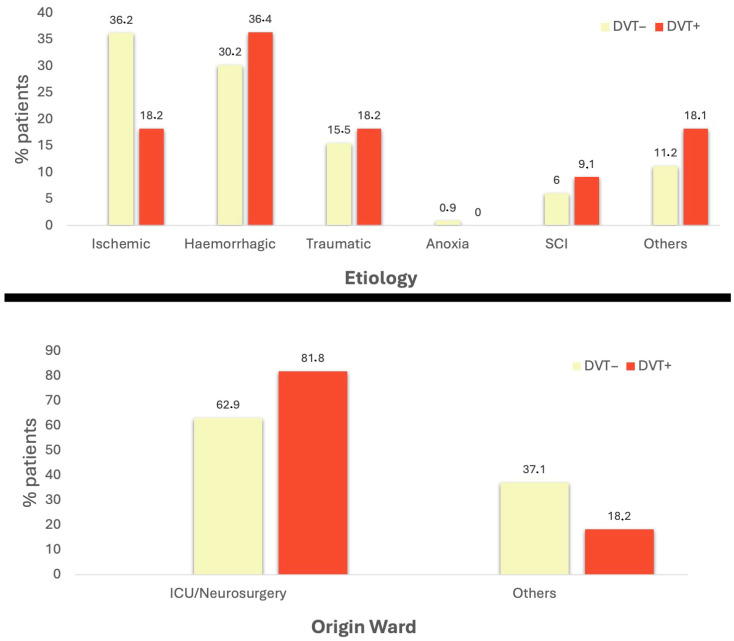
Etiology and origin ward prevalence in the ABI population based on DVT diagnosis. SCI: spinal cord injury; Others for Etiology: infections/benign tumor; Others for Origin Ward: Neurology, Psychiatry, and Geriatric.

**Table 1 healthcare-12-00936-t001:** Demographic and clinical characteristics of ABI patients on admission to IRU.

	DVT− (n° 116)	DVT+ (n° 11)	*p*-Level
Age	60.02 ± 17.8	61.45 ± 15.9	0.94
Male Sex, n (%)	72 (62.1%)	8 (72.7%)	0.48
Time from event (days)	30.8 ± 22.8	28.2 ± 11.8	0.72
**Risk Factors (before IRU admission)**			
Previous Motor Disability (% yes)	12 (10.3%)	0 (0%)	0.6
SEPSI (<1 month) (% yes)	29 (25%)	5 (45.5%)	0.16
Fractures (<3 months) (% yes)	21 (18.1%)	1 (9.1%)	0.69
Important Surgical Interventions (<3 months) (% yes)	51 (44%)	5 (45.4%)	1
**Clinical Evaluation**			
PADUA SCORE	4.7 ± 1.3	4.8 ± 1.07	0.54
Improve Bleeding Score	5.42 ± 2.66	6.4 ± 2.21	0.25
Improve Associative Score for VTE	4.21 ± 0.7	4.2 ± 0.6	0.86
Improve DD	5.6 ± 1.3	6 ± 0.9	0.27
Barthel Index	9 [0–75]	4 [0–20]	0.48
**Biochemical Data**			
D-Dimer	1.97 ± 1.61	6 ± 0.9	**0.001**
INR	1.24± 0.18	1.2 ± 0.09	0.96
PCR	3.31 ± 3.8	5.5 ± 4.2	**0.032**
PCT	0.5 ± 2.3	0.4 ± 0.6	0.94
VES	47.8 ± 34.7	66.7 ± 38.9	0.21
HB	11.7 ± 2.	11.7 ± 1.9	0.96
Leukocyte Count	8.9 ± 3.6	9.4 ± 2.8	0.44
PLT	291.9 ± 127.1	289.4 ± 76.6	0.56

Data are reported as %, median, or mean values accordingly. DVT: deep vein thrombosis. ICU: Intensive Care Unit. IRU: intensive rehabilitation unit. SCI: spinal cord injury. CNS: central nervous system. BOLD value means significant differences between groups at *p*-level < 0.05.

**Table 2 healthcare-12-00936-t002:** Pharmacological treatment of ABI patients after admission to IRU.

	DVT− (n° 116)	DVT+ (n° 11)	*p*-Level
Antiplatelet agents (% Yes)	30 (25.8%)	3 (27.3%)	0.91
Anticoagulation (% Yes)	11 (9.5%)	0 (0%)	0.97
TVB drug prevention			**0.04**
None	11 (9.5%)	0 (0%)	
ENOXAPARIN 2000 UI	1 (0.9%)	0 (0%)	
ENOXAPARIN 4000 UI	72 (62.1%)	5 (45.5%)	
ENOXAPARIN 6000 UI	16 (13.8%)	2 (18.2%)	
NADROPARIN 0.3 ML	2 (1.7%)	3 (27.3%)	
NADROPARIN 0.4 ML	7 (6%)	0 (0%)	
NADROPARIN 0.6 ML	4 (3.4%)	1 (9.1%)	
FONDAPARINUX 1.5 MG	0 (0%)	0 (0%)	
FONDAPARINUX 2.5 MG	3 (2.6%)	0 (0%)	

**Table 3 healthcare-12-00936-t003:** Clinical evolution during IRU period.

	DVT− (n° 116)	DVT+ (n° 11)	*p*-Level
N°	116	11	
Length of stay (days)	84.7 ± 58.4	96 ± 38.6	0.19
% of death during recovery	3/116 (2.5%)	1/11 (9%)	0.23
% of transfer to critical wards (yes)	17 (9.5%)	4 (36%)	0.058
Barthel Index at discharge	20 [0–95]	30 [0–100]	0.38

## Data Availability

The datasets associated with the present study are available upon reasonable request by interested researchers.

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
