# Peer review of "Prevalence and Risk Factors of Deep Venous Thrombosis in Intensive Inpatient Neurorehabilitation Unit"

_healthcare, 2024, doi:10.3390/healthcare12090936_

Round 1

Reviewer 1 Report

Comments and Suggestions for Authors

Dear Authors,

Thanks!

I would also like to add that my comments are always intended to contribute to the quality of the evaluated work.

In my opinion, the publication of this study requires a thorough revision of a large part of the sections reflected in the article. 

The study does not present an adequate rationale and grammar and scientific writing are also a problem. The writing is not “fluid” (please, see abstract, results and discussion).

Abstract:

Discussion ? Why ? Please, insert conclusion.

2.2. Procedure and Clinical Evaluations

This was a monocentric prospective observational study. At admission to the neu- 97 rorehabilitation unit, all patients underwent a detailed anamnesis and clinical evaluation. 98 Specifically, demographic data (sex, age, BMI), general, pharmacological, and DVT-spe- 99 cific anamnesis to evaluate the presence/absence of DVT possible risk factors, were col- 100 lected. The presence of DVT classic clinical signs, such as leg asymmetry with unilateral 101 edema, erythema, and pain, and the presence of Buer and Homans signs were evaluated. 102 Clinical evaluation at admission included also motor functional evaluation with the 103 Barthel Index and thrombosis/hemorrhagic risk evaluation with PADUA scores including 104 the improved score Bleeding, Improved Associative Score for VTE and improved DD 105 scales. Laboratory screening protocol at admission included: D-dimer quantification (nor- 106 mal value < 0,5 g/ml), complete blood count, creatinine, azotemia, electrolytes, reactive C 107 protein (RCP), erythrocyte sedimentation rate (ESR), PT, aPTT. All enrolled patients were 108 evaluated with a leg compressive ultrasonography at admission. Patients were clinically 109 daily evaluated for the presence of DVT for the entire duration of their rehabilitation stay” (please, insert studies/references).

2.3. Diagnosis of DVT

 Diagnosis of DVT was accomplished using an integrated clinical, biochemical and 112 instrumental evaluation. The DVT screening protocol consisted of a single quantitative D- 113 dimer assay using latex agglutination methods performed within 24 hours of rehabilita- 114 tion admission. The LIATEST D-dimer assay, which is an immuno-turbidimetric quanti- 115 tative assay method based on a latex microparticle agglutination test, was used, with a 116 positive D-dimer test for values > 0,5 g/ml. All venous duplex ultrasonography studies 117 were performed by the same radiographer with specific experience and reported by 118 trained radiologists using a Canon Aplio 300 ultrasound machine during the entire study 119 period. Compressive ultrasound was performed on each patient within 72 hours of ad- 120 mission. The presence of DVT was determined by a positive venous du-plex ultrasonog- 121 raphy result. Each patient without a diagnosis of DVT and admission time was clinically 122 followed for the entire length of stay in the rehabilitation yard for DVT symptoms and 123 signs of early detection. In case of clinical suspicion, D-dimer testing and CUS were per- 124 formed. The diagnosis of PE was made on clinical grounds and investigated accord- 125 ingly. (please, insert studies/references).

2.4. Statistical Analysis

“Statistical analysis was performed using IBM SPSS software (version 26; Statistical Pack- 128 age for Social Sciences; www.spss.it). Please, insert IBM.

Table 1. Demographic and clinical characteristics of ABI patients at admission in IRU.

Table 2. Clinical evolution during IRU period.

N = population: n = sample. Please, insert n.

Discussion

The organization of the theoretical framework seems insufficient when it comes to constructing the discussion of the study,

A summary of the most relevant results be appreciated in the first paragraph.

It would also be useful to establish some kind of division within the discussion, according to the objectives addressed. 

Conclusion

Please, insert: Practical Implications, Limitations, and Suggestions for Future Research

References

Please, insert recent studies (2023/2024).

Thanks

Kind regards

Comments on the Quality of English Language

Author Response

Reviewer n°1

I would also like to add that my comments are always intended to contribute to the quality of the evaluated work.  In my opinion, the publication of this study requires a thorough revision of a large part of the sections reflected in the article. 

The study does not present an adequate rationale and grammar and scientific writing are also a problem. The writing is not “fluid” (please, see abstract, results and discussion).

REPLY: We would like to express our appreciation for the reviewer’s comments. We feel that our manuscript is strongly improved by incorporating these suggestions. In the attached reply to the reviewer, we outlined our responses to each of his/her comments. 

1) Abstract: Discussion ? Why ? Please, insert

REPLY: The abstract has been Modified

2) Procedure and Clinical Evaluations: This was a monocentric prospective observational study. At admission to the neu- 97 rorehabilitation unit, all patients underwent a detailed anamnesis and clinical evaluation. 98 Specifically, demographic data (sex, age, BMI), general, pharmacological, and DVT-spe- 99 cific anamnesis to evaluate the presence/absence of DVT possible risk factors, were col- 100 lected. The presence of DVT classic clinical signs, such as leg asymmetry with unilateral 101 edema, erythema, and pain, and the presence of Buer and Homans signs were evaluated. 102 Clinical evaluation at admission included also motor functional evaluation with the 103 Barthel Index and thrombosis/hemorrhagic risk evaluation with PADUA scores including 104 the improved score Bleeding, Improved Associative Score for VTE and improved DD 105 scales. Laboratory screening protocol at admission included: D-dimer quantification (nor- 106 mal value < 0,5 g/ml), complete blood count, creatinine, azotemia, electrolytes, reactive C 107 protein (RCP), erythrocyte sedimentation rate (ESR), PT, aPTT. All enrolled patients were 108 evaluated with a leg compressive ultrasonography at admission. Patients were clinically 109 daily evaluated for the presence of DVT for the entire duration of their rehabilitation stay” (please, insert studies/references).

REPLY: Done

3) Diagnosis of DVT: Diagnosis of DVT was accomplished using an integrated clinical, biochemical and 112 instrumental evaluation. The DVT screening protocol consisted of a single quantitative D- 113 dimer assay using latex agglutination methods performed within 24 hours of rehabilita- 114 tion admission. The LIATEST D-dimer assay, which is an immuno-turbidimetric quanti- 115 tative assay method based on a latex microparticle agglutination test, was used, with a 116 positive D-dimer test for values > 0,5 g/ml. All venous duplex ultrasonography studies 117 were performed by the same radiographer with specific experience and reported by 118 trained radiologists using a Canon Aplio 300 ultrasound machine during the entire study 119 period. Compressive ultrasound was performed on each patient within 72 hours of ad- 120 mission. The presence of DVT was determined by a positive venous du-plex ultrasonog- 121 raphy result. Each patient without a diagnosis of DVT and admission time was clinically 122 followed for the entire length of stay in the rehabilitation yard for DVT symptoms and 123 signs of early detection. In case of clinical suspicion, D-dimer testing and CUS were per- 124 formed. The diagnosis of PE was made on clinical grounds and investigated accord- 125 ingly. (please, insert studies/references).

REPLY: Done

4) Statistical Analysis: “Statistical analysis was performed using IBM SPSS software (version 26; Statistical Pack- 128 age for Social Sciences; www.spss.it). Please, insert IBM.

REPLY: Modified

5) Table 1. Demographic and clinical characteristics of ABI patients at admission in IRU. Table 2. Clinical evolution during IRU period. N = population: n = sample. Please, insert n.

REPLY: Done

6) Discussion: The organization of the theoretical framework seems insufficient when it comes to constructing the discussion of the study, A summary of the most relevant results be appreciated in the first paragraph. It would also be useful to establish some kind of division within the discussion, according to the objectives addressed. 

REPLY: The discussion section has been modified according to the reviewer’s suggestion  

7) Conclusion: Please, insert: Practical Implications, Limitations, and Suggestions for Future Research

REPLY: Done

8) References: Please, insert recent studies (2023/2024).

REPLY: The entire reference section has been reformulated adding new recent studies

Reviewer 2 Report

Comments and Suggestions for Authors

Prevalence and Risk Factors of Deep Venous Thrombosis in Intensive Inpatient Neuro-Rehabilitation

Recommendation: Major Changes for Revise Submission

In this paper the authors assess the effect of prevalence and risk factors of deep venous thrombosis in intensive inpatient neuro-rehabilitation. The study's hypothesis is less compelling because it has a many significant flaws that need to be fixed and more material in the publication could help to improve the paper's clarity of presentation. Therefore, this reviewer considers that the manuscript does have some important drawbacks that are listed below.

1.     The abstract is not written in a formal style that accurately conveys the manuscript's technical soundness. The abstract should be revised by the authors in accordance with the journal format to better represent the manuscript's technical soundness.

2.     The real problem statement, which explains why the authors' investigation and suggested conclusions are required, has not been explained by the authors in detail. I haven't noticed it in the introduction or abstract.

3.     It would be also interesting to know more about "ABI patients were screened for DVT", explanation are missing from the paper. More detail is necessary.

The research hypotheses are not clear in the manuscript. Based on research hypothesis it should be mentioned in the introduction section clearly the research hypothesis that what the authors analyze via this research.

The authors should get feedback from field expert (Neuro-Surgeons) via subjective questionnaires and include in the manuscript as a separate section that what suggests by expert in the relevant context. So that, based on their suggestions then the authors propose their idea.

The most important thing is the lack of literature review; the authors have not included the literature review. Based on literature review it will be clear the gap in the existing techniques.

The methodology is not clear. It is necessary to include a flow diagram regarding the proposed methodology and describe the work in a specific way.

The conclusion section has not been answered to fully evaluate the effectiveness and potential impact of the work as a whole that “do the chosen methodologies produce good results? How do the chosen methodologies affect the overall study basis on comparison with other existing methodologies?” Answer of this question in a proper statement is mandatory in conclusion section.

s: In my opinion, it is also necessary’s to show some latest references to healthcare (i.e., 2022-2024), since there is no reference about this technology. In the same way, it would be good to take the previous literature analyzed about neuro-rehabilitation.

Table 1 should split in multiple tables and it would be much easier to interpret the result if Tables 1 and 2 are given as a graph (e.g, histogram).

Comments on the Quality of English Language

See my attached comments

Author Response

Reviewer n°2

In this paper the authors assess the effect of prevalence and risk factors of deep venous thrombosis in intensive inpatient neuro-rehabilitation. The study's hypothesis is less compelling because it has a many significant flaws that need to be fixed and more material in the publication could help to improve the paper's clarity of presentation. Therefore, this reviewer considers that the manuscript does have some important drawbacks that are listed below.

REPLY: We would like to express our appreciation for the reviewer’s comments. We feel that our manuscript is strongly improved by incorporating these suggestions. In the attached reply to the reviewer, we outlined our responses to each of his/her comments. 

  1. The abstract is not written in a formal style that accurately conveys the manuscript's technical soundness. The abstract should be revised by the authors in accordance with the journal format to better represent the manuscript's technical soundness.

REPLY: The abstract has been re-formulated according to the reviewer’s suggestion.

  1. The real problem statement, which explains why the authors' investigation and suggested conclusions are required, has not been explained by the authors in detail. I haven't noticed it in the introduction or abstract.

REPLY: The introduction has completely been re-formulated according to the reviewer’s suggestion.

  1. It would be also interesting to know more about "ABI patients were screened for DVT", explanation are missing from the paper. More detail is necessary.

REPLY: This section has been modified following the reviewer’s suggestion.

4) The research hypotheses are not clear in the manuscript. Based on research hypothesis it should be mentioned in the introduction section clearly the research hypothesis that what the authors analyze via this research.

REPLY: The research hypothesis in the abstract and introduction has completely been re-formulated according to the reviewer’s suggestion.

5) The authors should get feedback from field expert (Neuro-Surgeons) via subjective questionnaires and include in the manuscript as a separate section that what suggests by expert in the relevant context. So that, based on their suggestions then the authors propose their idea.

REPLY: In this study, we did not collect neurosurgical opinions. It has been demonstrated that patients who developed VTE were more likely to have undergone placement of invasive intracranial monitoring or neurosurgical intervention via craniotomy/craniectomy than their counterparts who did not develop VTE (Cole et al., Neurocrit Care (2024)). We included this limitation in the discussion section.

6) The most important thing is the lack of literature review; the authors have not included the literature review. Based on literature review it will be clear the gap in the existing techniques.

REPLY: The entire reference section and literature have been reformulated adding new recent studies

7) The methodology is not clear. It is necessary to include a flow diagram regarding the proposed methodology and describe the work in a specific way.

REPLY: A new figure has been included to better explain our procedural phases.

8) The conclusion section has not been answered to fully evaluate the effectiveness and potential impact of the work as a whole that “do the chosen methodologies produce good results? How do the chosen methodologies affect the overall study basis on comparison with other existing methodologies?” Answer of this question in a proper statement is mandatory in conclusion section.

REPLY: This suggestion has been included in the conclusion. 

  • In my opinion, it is also necessary’s to show some latest references to healthcare (i.e., 2022-2024), since there is no reference about this technology. In the same way, it would be good to take the previous literature analyzed about neuro-rehabilitation.

REPLY: As said before the entire reference section and literature have been reformulated adding new recent studies

10) Table 1 should split in multiple tables and it would be much easier to interpret the result if Tables 1 and 2 are given as a graph (e.g, histogram).

REPLY: Done. A new table 2, and a new figure have been included.

Reviewer 3 Report

Comments and Suggestions for Authors

The article, entitled "Prevalence and Risk Factors of Deep Venous Thrombosis in Intensive Inpatient Neuro-Rehabilitation," highlights that deep venous thrombosis (DVT) prevalence was approximately 8.6% in Acquired Brain Injury (ABI) patients. The study found that D-dimer levels were significantly higher in patients with DVT, and there was a notable difference in pharmacological DVT prophylaxis usage and the prevalence of transfers in critical wards among these patients.

Regarding your points:

  1. It would be beneficial to know the specific criteria for ABI patients who received DVT prophylaxis, particularly considering that some ABI patients did not utilize pharmaceutical prophylaxis.
  2. The observation that ABI patients with DVT are more likely to receive DVT prophylaxis than those without DVT raises questions due to the seemingly paradoxical nature of this finding. The small sample size of ABI patients with DVT (n=11) might have influenced the results, potentially leading to a counterintuitive outcome. An explanation for this unexpected result would be valuable.
  3. It is recommended to perform additional analyses, such as Cox regression or logistic regression, to identify predictors of DVT occurrence and to present findings through more tables or graphs for a more comprehensive understanding.
  4. I suggest adding a section in the discussion highlighting the limitations of the study to provide a more well-rounded interpretation of the results.
  5. To enhance clarity, it's advisable to condense the conclusion section for a more concise summary of the study's key findings.

Author Response

The article, entitled "Prevalence and Risk Factors of Deep Venous Thrombosis in Intensive Inpatient Neuro-Rehabilitation," highlights that deep venous thrombosis (DVT) prevalence was approximately 8.6% in Acquired Brain Injury (ABI) patients. The study found that D-dimer levels were significantly higher in patients with DVT, and there was a notable difference in pharmacological DVT prophylaxis usage and the prevalence of transfers in critical wards among these patients.

REPLY: We would like to express our appreciation for the reviewer’s comments. We feel that our manuscript is strongly improved by incorporating these suggestions. In the attached reply to the reviewer, we outlined our responses to each of his/her comments. 

Regarding your points:

  1. It would be beneficial to know the specific criteria for ABI patients who received DVT prophylaxis, particularly considering that some ABI patients did not utilize pharmaceutical prophylaxis.

REPLY: In this study, we reported the therapy that each patient was taking at rehabilitation admission time. The therapy that rehabilitation clinicians prescribe after patient clinical evaluation is not analyzed here, because is not in line with the aim of the study. For this reason, we did not specify clinical criteria to justify why some patients were not treated with thromboprophylaxis drugs.

2) The observation that ABI patients with DVT are more likely to receive DVT prophylaxis than those without DVT raises questions due to the seemingly paradoxical nature of this finding. The small sample size of ABI patients with DVT (n=11) might have influenced the results, potentially leading to a counterintuitive outcome. An explanation for this unexpected result would be valuable.

REPLY: As suggested, we added some comments in the text, Discussion section.

3) It is recommended to perform additional analyses, such as Cox regression or logistic regression, to identify predictors of DVT occurrence and to present findings through more tables or graphs for a more comprehensive understanding.

REPLY: We appreciate the reviewer's recommendation to run further analyses, including logistic or Cox regression. Although this analysis was already taken into consideration in the previous edition, parameter estimates can become unstable when there is a small sample size (n=11). In light of these factors, we have chosen not to do regression analysis for this study to avoid producing an incorrect result.

4) I suggest adding a section in the discussion highlighting the limitations of the study to provide a more well-rounded interpretation of the results

REPLY: Done  

5) To enhance clarity, it's advisable to condense the conclusion section for a more concise summary of the study's key findings.

REPLY: Done

Round 2

Reviewer 1 Report

Comments and Suggestions for Authors

Dear Authors,

Great job!

Thanks!

Kind regards

Comments on the Quality of English Language

-

Author Response

Thanks a lot !!

Reviewer 2 Report

Comments and Suggestions for Authors

Comments:

This is the second round of this manuscript and this reviewer have read the manuscript and found that the authors have been addressed the changes carefully given by this reviewer and the manuscript has significantly improved. However, after evaluation of all changes in the manuscript this reviewer is willing to accept the manuscript for publication in Health Care Journal. I appreciate the efforts of all authors during this research. In addition, the following are minor changes for the further improvement of this manuscript:

1.     Further some latest relevant references are necessary to add in the manuscript regarding 2024 published articles.

2.     In section 2.2, write effective bullet points to achieve this sparingly and strategically. In current form all the points are not well attractive for readers.

3.     Figure 1 is still not designed according to the correct flow of work. For example, connect DVT+ and DVT- as separate boxes with IRU admission box.

4.     In Figure 2, the vertical values show the percentage sign (%) but it requires meaningful words such as performance or efficiency in percentage as a vertical label. In addition also add the percentage actual values on each bar.

5.     In conclusion, the authors have still not concluded that the chosen methodologies produce good results. The real values of the outcome must be mentioned in a brief statement.

6.     Finally, the limitations section should be merged with conclusion section as a future direction.

Comments on the Quality of English Language

Proofread the paper by a native or English expert.

Author Response

  1. Further some latest relevant references are necessary to add in the manuscript regarding 2024 published articles.

REPLY: Done. See the conclusion section

2. In section 2.2, write effective bullet points to achieve this sparingly and strategically. In current form all the points are not well attractive for readers.

REPLY: the section 2.2 has been completely re-formulated.

3. Figure 1 is still not designed according to the correct flow of work. For example, connect DVT+ and DVT- as separate boxes with IRU admission box.

REPLY: Figure 1 has been modified accordingly.

4. In Figure 2, the vertical values show the percentage sign (%) but it requires meaningful words such as performance or efficiency in percentage as a vertical label. In addition, also add the percentage actual values on each bar.

REPLY: Figure 2 has been modified accordingly.

5. In conclusion, the authors have still not concluded that the chosen methodologies produce good results. The real values of the outcome must be mentioned in a brief statement.

REPLY: Conclusion has been modified accordingly.  

6. Finally, the limitations section should be merged with the conclusion section as a future direction.

REPLY: DONE

Reviewer 3 Report

Comments and Suggestions for Authors

The authors have answered the reviewer’s points accordingly. Therefore, I suggest accepting the manuscript after minor revision.

1.         Results section:

Overall, the pharmacological treatment for DVT+ patients differed from that of DVT- patients since they were more frequently prescribed Nadroparin 0.3 ml for thrombosis prophylaxis (27.3% vs 1.7%, p value= 0.04)

Author Response

Results section:

Overall, the pharmacological treatment for DVT+ patients differed from that of DVT- patients since they were more frequently prescribed Nadroparin 0.3 ml for thrombosis prophylaxis (27.3% vs 1.7%, p value= 0.04)

REPLY: Done